# Pirfenidone Inhibits Alveolar Bone Loss in Ligature-Induced Periodontitis by Suppressing the NF-κB Signaling Pathway in Mice

**DOI:** 10.3390/ijms24108682

**Published:** 2023-05-12

**Authors:** Zijiao Zhang, Juhan Song, Seung-Hee Kwon, Zhao Wang, Suk-Gyun Park, Xianyu Piao, Je-Hwang Ryu, Nacksung Kim, Ok-Su Kim, Sun-Hun Kim, Jeong-Tae Koh

**Affiliations:** 1Department of Pharmacology and Dental Therapeutics, School of Dentistry, Chonnam National University, Gwangju 61186, Republic of Korea; 2Hard-Tissue Biointerface Research Center, School of Dentistry, Chonnam National University, Gwangju 61186, Republic of Korea; 3Department of Pharmacology, Chonnam National University Medical School, Gwangju 61469, Republic of Korea; 4Department of Periodontology, School of Dentistry, Chonnam National University, Gwangju 61186, Republic of Korea; 5Department of Oral Anatomy, School of Dentistry, Chonnam National University, Gwangju 61186, Republic of Korea

**Keywords:** pirfenidone, periodontitis, osteoclast differentiation, alveolar bone loss, inflammation, NF-κB pathway

## Abstract

There has been increasing interest in adjunctive use of anti-inflammatory drugs to control periodontitis. This study was performed to examine the effects of pirfenidone (PFD) on alveolar bone loss in ligature-induced periodontitis in mice and identify the relevant mechanisms. Experimental periodontitis was established by ligating the unilateral maxillary second molar for 7 days in mice (n = 8 per group), and PFD was administered daily via intraperitoneal injection. The micro-computed tomography and histology analyses were performed to determine changes in the alveolar bone following the PFD administration. For in vitro analysis, bone marrow macrophages (BMMs) were isolated from mice and cultured with PFD in the presence of RANKL or LPS. The effectiveness of PFD on osteoclastogenesis, inflammatory cytokine expression, and NF-κB activation was determined with RT-PCR, Western blot, and immunofluorescence analyses. PFD treatment significantly inhibited the ligature-induced alveolar bone loss, with decreases in TRAP-positive osteoclasts and expression of inflammatory cytokines in mice. In cultured BMM cells, PFD also inhibited RANKL-induced osteoclast differentiation and LPS-induced proinflammatory cytokine (IL-1β, IL-6, TNF-a) expression via suppressing the NF-κB signal pathway. These results suggest that PFD can suppress periodontitis progression by inhibiting osteoclastogenesis and inflammatory cytokine production via inhibiting the NF-κB signal pathway, and it may be a promising candidate for controlling periodontitis.

## 1. Introduction

Periodontitis is a chronic inflammatory disease caused by dysbiotic microbiota and host immune responses, which can lead to the progressive destruction of the tooth-supporting apparatus [1]. The traditional treatment for periodontitis involves surgical and non-surgical approaches based on eliminating the pathogenic biofilm. Recently, the adjunctive use of anti-inflammatory drugs for suppressing the host immune response has also been considered to control periodontitis.

Periodontitis has two major hallmarks: inflammatory cytokine production in the periodontium and alveolar bone loss [2,3]. In periodontitis, the expression of inflammatory cytokines, such as interleukin-1 beta (IL-1β), interleukin-6 (IL-6), and tumor necrosis factor-alpha (TNF-a), is increased, and it is closely related to the destruction of periodontal tissue and alveolar bone loss [4]. Additionally, the increase in these inflammatory cytokines enhances the expression of matrix metalloproteinases (MMPs) involved in tissue destruction and the expression of nuclear factor kappa light-chain enhancer of activated B cells (NF-κB) and receptor activator of nuclear factor kappa-B ligand (RANKL), which promote inflammation and osteoclast differentiation [5,6]. Control of inflammatory responses in the host tissue is necessary to treat periodontitis.

The bone tissue, including alveolar bone, is maintained by the balance between osteoblasts and osteoclasts [7]. The alveolar bone loss in periodontitis results from an imbalance between osteoblasts and osteoclasts: increased osteoclast activity [3,8]. Osteoclasts are derived from monocyte/macrophage lineage cells, and their differentiation is promoted by the action of RANKL and inflammatory cytokines [5]. RANKL binds to its RANK receptor on the cell membrane, increasing the activity of nuclear factor of activated T-cells 1 (NFATc1) and c-Fos through the NF-κB and mitogen-activated protein kinases (MAPK) signaling pathways, subsequently increasing the expression of cathepsin K (CtsK), tartrate-resistant acid phosphatase (TRAP), MMPs, and so forth [9]. Therefore, osteoclast activity increases, and calcified matrices are resorbed. A few inflammatory cytokines, including IL-1β, enhance the action of RANKL on osteoclasts [5]. Therefore, strategies of suppressing osteoclast differentiation have been considered for restoring alveolar bone loss as a result of periodontitis.

Pirfenidone (5-methyl-1-phenyl-2(1H)-pyridone, PFD) is a synthetic derivative of pyridine that is orally administered, rapidly absorbed, and has high stability in the body. Currently, it is approved by the Food and Drug Administration for treating idiopathic pulmonary fibrosis (IPF) [10]. The exact mechanism of action of PFD is not fully understood, but it reportedly reduces inflammation by suppressing the NF-κB pathway and oxidative stress [11,12]. The use of PFD for treating pulmonic injury and nonalcoholic steatohepatitis inhibits the expression of proinflammatory cytokines, including IL-1β, IL-6 and TNF-a, and MMPs [13,14,15]. We can, therefore, speculate that PFD may be useful for controlling periodontitis.

In this study, we evaluated the therapeutic potential of PFD in ligature-induced periodontitis in mice, and we determined the effects of PFD on the osteoclast differentiation and expression of inflammatory cytokines with molecular mechanisms in vitro.

## 2. Results

### 2.1. PFD Inhibits Ligature-Induced Alveolar Bone Loss in Mice

To examine the effects of PFD on periodontitis, ligature-induced periodontitis was developed around the maxillary second molar in right side of the mouth in mice, and phosphate buffer saline (PBS) or PFD (60 mg/kg, 120 mg/kg) was injected peritoneally, as shown Figure 1A. During the experiment, there were no significant changes in body weight due to treatment with PFD (Figure 1B). The micro-computed tomography (μ-CT) and histology results (Figure 1C,D) showed that the height, density, and thickness of alveolar bone surrounding silk-ligated second molars were remarkably decreased compared with those in the control sides. Moreover, the silk ligation induced the increases in distance between the cementoenamel junction and alveolar bone crest (CEJ-ABC distance) and the number of TRAP-positive osteoclasts compared with the control sides. In contrast, the ligature-induced alveolar bone loss was inhibited by PFD treatment (Figure 1C–E). Quantitative analyses support the findings that treatment with PFD dose-dependently inhibited the ligature-induced alteration in CEJ-ABC distance, bone volume per tissue volume (BV/TV), bone mineral density (BMD), and TRAP-positive osteoclasts to control level. However, PFD did not produce any significant alteration in the control-side alveolar bone without silk ligation (Figure 1D,E).

### 2.2. PFD Suppresses RANKL-Induced Osteoclast Differentiation in Bone Marrow Macrophages (BMMs)

PFD treatment decreased the number of ligature-induced TRAP-positive cells in mice (Figure 1C,E). Thereby, we further examined the effects of PFD on RANKL-induced osteo-clastogenesis in BMM cell cultures. PFD has no cytotoxicity on BMMs up to 800 µM (Figure 2A). Treatment with PFD dose-dependently decreased the number of TRAP-positive cells, the size of F-actin belts, and the area of resorption pits on bone slices under exposure of RANKL and M-CSF, indicating that PFD inhibited RANKL-induced osteoclast differentiation of BMMs (Figure 2B–D). In addition, RNA isolation real-time polymerase chain reaction (RT-PCR) analysis demonstrated that PFD dose-dependently inhibited the osteoclast-specific gene expressions, including *c-Fos*, *Nfatc1*, *Ctsk*, *Trap*, and *Mmp9 mRNA*, under stimuli of RANKL and M-CSF (Figure 2E). Western blotting also revealed that PFD treatment suppressed the protein expression of osteoclast-specific genes (Figure 2F,G).

### 2.3. PFD Suppresses RANKL-Induced Osteoclast Differentiation by Inhibiting the NF-κB Pathway

RANKL activates the NF-κB and MAPK signaling pathways to stimulate osteoclast differentiation [16]. In this study, RANKL treatment also increased the NF-κB pathways in BMMs (increases in phosphorylation of IκB kinase alpha/beta (IKKa/β), p65, and nuclear factor of kappa light polypeptide gene enhancer in B-cell inhibitor alpha (IκBa)). PFD treatment dose-dependently inhibited the RANKL-induced phosphorylation of proteins (Figure 3A,B). However, PFD treatment did not affect the RANKL activation of MAPK pathways, phosphorylation of p38, c-Jun N-terminal kinase (JNK), and extracellular signal-regulated kinases (ERKs) (Appendix A).

The NF-κB (p65) translocation into the nucleus activates target gene expression [17,18]. Considering that PFD inhibited RANKL activation of the NF-κB pathway (Figure 3A,B), we hypothesized that PFD treatment may also inhibit the NF-κB (p65) nuclear translocation. In Figure 3C,D, RANKL treatment increased the nuclear fraction of p65 protein, and PFD treatment significantly decreased the level of nuclear p65 in a dose-dependent fashion. The immunofluorescence results consistently confirm that PFD inhibited the nuclear translocation of p65 (Figure 3E).

**Figure 3 ijms-24-08682-f003:**
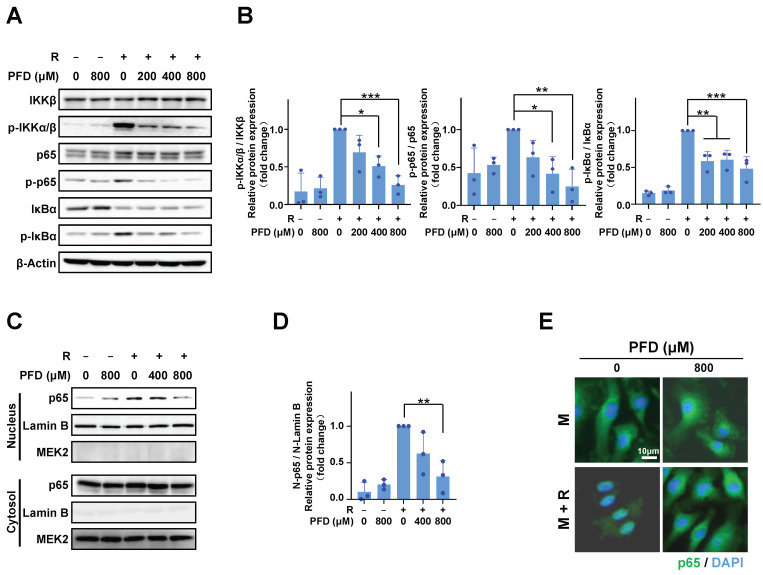
PFD inhibits RANKL-induced NF-κB pathway and p65 translocation to nuclear in BMMs. (**A**) A representative Western blot image. (**B**) A relative gray densitometric analysis of (**A**). The fold value was based on RANKL group. (**C**) Representative Western blot images of the effects of PFD on RANKL-induced p65 translocation to nucleus. Lamin B and MEK2 were regarded as markers for nuclear and cytosol protein expression, respectively. (**D**) The relative gray densitometric analysis of p65 expression in nucleus. The fold value was based on RANKL group. All quantifications of images were measured in Image J Fiji (Version 1.2) from three independent experiments, respectively. (**E**) Confocal images of p65 (green) translocation into nucleus (DAPI, blue) after PFD treatment for 30 min. Scale bar, 10 µm. Pirfenidone, PFD; M, MCSF (30 ng/mL); R, RANKL (100 ng/mL); N, nucleus. All bar graphs are showed as the mean ± SD. * *p* < 0.05, ** *p* < 0.01, and *** *p* < 0.001 compared with the RANKL group.

### 2.4. PFD Did Not Affect Osteoblast Differentiation of Mesenchymal Stem Cells (BMSCs)

Bone homeostasis is maintained by osteoblasts cooperating with osteoclasts; therefore, we also examined the effects of PFD on osteoblast differentiation of BMSCs. PFD has no cytotoxicity on BMSCs up to 800 µM (Appendix A). When BMSCs were exposed to an osteogenic medium (OM), calcium deposition (alizarin red staining, ARS; Appendix A) and osteoblast-specific gene expression (*alkaline phosphatase*, *Alp*; *osterix*, *Osx*; *runt-related transcription factor 2*, *Runx2*; Appendix A) were increased. However, PFD treatment did not affect the potential of osteoblastic differentiation from BMSCs (Appendix A).

### 2.5. PFD Suppresses Proinflammatory Cytokine Expression by Inhibiting the NF-κB Signaling Pathway

Histology and immunohistochemistry (IHC) staining results showed that silk ligation surrounding maxillary second molar increased the loss of alveolar bone with the increases in IL-1β, IL-6, and TNF-a expression, and intraperitoneal administration of PFD dose-relatedly decreased the expression on the periodontitis side (Figure 4). In the control side without silk ligation, there were no significant differences in the proinflammatory cytokine expression between control PBS- and PFD-treated groups (Figure 4).

In BMM cultures, lipopolysaccharide (LPS) treatment significantly increased *Il-1β*, *Il-6*, and *Tnf-a* mRNA expression and PFD treatment inhibited the expression dose-dependently (Figure 5A). Concordantly, Western blot analysis also showed that PFD treatment inhibited the LPS induction of pro-IL-1β, IL-1β, and IL-6 expression dose-dependently (Figure 5B,C). In addition, PFD inhibited the LPS-induced phosphorylation of p65 and IκBa (Figure 5D,E). A schematic representation of the PFD mechanism model in periodontitis is shown in Figure 5F.

## 3. Discussion

Recently, there has been increasing interest in repurposing existing drugs for dentistry [19]. In this study, we found that PFD for IPF inhibits alveolar bone loss and proinflammatory cytokine expression in ligature-induced periodontitis in mice. Furthermore, the protective effects of PFD resulted from inhibiting osteoclast differentiation and inflammatory cytokine expression via suppressing the NF-κB pathway.

PFD is widely used for the treatment of idiopathic pulmonary fibrosis due to its significant anti-inflammatory and antifibrotic properties, which inhibit the activity of several cytokines and growth factors in lung tissue damage [19,20]. The recommended oral dose for the treatment of IPF patients is 1800 mg/kg [20], and side effects, such as nausea, diarrhea, skin rash, and salivation, have been reported [21]. A previous study showed that 125 mg/kg/ip/day of PFD has antifibrotic effectiveness in mouse liver fibrosis without causing any mortality, although it was associated with minimal side effects [22]. In the present study, we evaluated the efficacy of PFD at doses of 60 and 120 mg/kg/ip/day for the treatment of ligature-induced periodontitis. The doses of PFD showed therapeutic effectiveness without causing any significant systemic adverse effects.

Alveolar bone loss in periodontitis is closely related to an increased osteoclast activity. Osteoclasts originate from hematopoietic stem cells, and RANKL derived from osteoblasts or immune cells promotes their differentiation. RANKL activates the RANKL/RANK/TRAF6 pathways and its downstream NF-κB, MAPK, or PI3K/Akt pathways [23,24] to induce the expression of osteoclast-specific genes (Nfatc1, Ctsk, Trap, Mmp9) and osteoclastogenesis [25,26]. Bacterial LPS enhances the RANKL-induced osteoclast differentiation and activity [25]. In addition, LPS promotes survival of RANKL-induced mature osteoclasts by activating Akt, NF-κB, and ERK via MyD88 and producing TNF-a [27,28,29]. In this study, silk ligature induced alveolar bone loss along with an increase in TRAP-positive osteoclasts, and administration of PFD mitigated this effect. In BMM cultures, RANKL treatment increased osteoclast-specific gene expression and the number of TRAP-positive multinucleated cells, F-acting ring formation, and bone resorption pits. Furthermore, PFD administration induced a decrease in gene expression along with suppression of osteoclast activity. These results indicate that PFD can directly inhibit osteoclast differentiation to suppress alveolar bone loss in periodontitis.

The bacterial LPS stimulates the host’s immune response to produce various proinflammatory cytokines as well as RANKL expression in many tissues, including PDL cells. When LPS was exposed to human airway epithelial cells and macrophages, expressions of several proinflammatory cytokines increased via NF-κB, signal transducer, and activator of transcription 3 (STAT3) or activator protein-1 (AP-1) [30]. In another study, PFD treatment attenuated LPS-activated inflammatory responses in lung injury [13]. In the present study, LPS treatment also increased TNF-a, IL-1β, and IL-6 expression in BMMs, similar to those in lung cells, and PFD inhibited the LPS induction of cytokines. These results consistently support that PFD may suppress the pathogenesis of periodontitis via simultaneously controlling the osteoclastogenesis and inflammatory cytokine expression. However, PFD does not appear to have a significant role in osteoblastic differentiation from BMSCs (Appendix A).

The precise molecular mechanism of PFD is not fully understood, but it reportedly suppresses inflammation by inhibiting the transcription factor NF-κB [31,32]. Functionally, the NF-κB pathway serves as a critical mediator of inflammatory responses and osteoclastogenesis [27,33]. PFD suppressed LPS-induced TNF-a, IL-1β, and IL-6 expression through inhibiting NF-κB activation in RAW264.7 cells, and it also ameliorated the activation of NF-κB signaling in the livers of mice fed a high-fat diet [25]. Our results showed that PFD could suppress the RANKL- or LPS-induced NF-κB activation and p65 translocation into the nuclei in BMMs. However, PFD did not affect the MAPK pathway, which is another important pathway for osteoclastogenesis. These findings suggest that PFD can alleviate periodontal inflammation and the subsequent alveolar bone loss via suppressing the NF-κB pathway.

Overall, our results show that PFD is useful for regulating periodontitis. However, there are still some limitations to directly using PFD for treating periodontitis in humans, due to non-validation of its human effects. It is necessary to confirm whether periodontal symptoms can be relieved in patients who have taken PFD for IPF.

## 4. Materials and Methods

### 4.1. Drugs

The PFD (purity ≥ 98%) was obtained from Aladdin Biochemical technology, Co. (Shanghai, China) and diluted in PBS for in vivo and in vitro experiments.

### 4.2. Cell Viability Assay

The cell viability was assessed using the 3-(4,5-Dimethylthiazol-2-yl)-2,5-Diphenyltetrazolium Bromide (MTT, Sigma Aldrich, St. Louis, MO, USA) assay reagent after treating with compatible doses of pirfenidone (PFD). Bone marrow macrophages (BMMs) were seeded at a density of 4 × 10^5^/mL in a growth medium (GM) with M-CSF (30 ng/mL) in 96-well plates overnight. The GM was α-MEM (Gibco, Carlsbad, CA, USA) with fetal bovine serum (FBS, 10%, Gibco, Carlsbad, CA, USA) and penicillin/streptomycin (P/S, 1%, Gibco, Carlsbad, CA, USA). Different doses of PFD (0, 100, 200, 400, 800 µM) were then added to the GM, and the samples were incubated for 24 h. The MTT reagent (0.5 mg/mL) was added to each well and cultured at 37 °C for 2 h. The purple formazan crystals indicated complete solubilization by dimethyl sulfoxide (DMSO, Sigma Aldrich, St. Louis, MO, USA), and the absorbance of the samples was measured by using a microplate reader (Multiskan Sky High Microplate Reader, Thermo Fisher, CA, USA) at 562 nm. The cell viability of the mesenchymal stem cells was also analyzed using the MTT assay method and the same protocol with a different cell density (6 × 10^4^/mL) in the GM for 2 and 4 days.

### 4.3. Animal Experiments

Eight-week-old C57BL/6 male mice were purchased from Damool Science (Daejeon, Republic of Korea). All animal studies were approved by the guidelines of Animal Care and Use Committee of Chonnam National University. The 24 mice were randomly divided into 3 groups (*n* = 8 per group), and PFD (60 mg/kg or 120 mg/kg) or PBS was intraperitoneally injected once a day for 8 days from one day before silk ligation. Experimental periodontitis in mice was produced via silk ligature method; the maxillary right second molar was ligated with 0–5 silks for 7 days (Abe and Hajishengallis, 2013). The left second molar without ligation was used as the control in each mouse.

### 4.4. μ-CT Analysis

The microarchitectural properties of alveolar bone tissues were imported with μ-CT system (model 1172, Skyscan, Aartselaar, Belgium), with the scanning parameters set to 50 kV and 200 μA. The images were reconstructed using NRecon software (Version 1.7.4.2) and mimics imaging program (version 14.0, Materialise N.V., Leuven, Belgium) [34]. The results of BV/TV, BMD, and CEJ-ABC were obtained within the region of interest, which surrounds the second molar [35,36].

### 4.5. IHC Analyses

The maxillary tissues were fixed with 10% formalin (Sigma Aldrich, USA), decalcified with 0.5 M ethylene-diamine tetra acetic acid (EDTA, pH 7.4), and paraffin-embedded. The specimens were sliced into 5 µm thick sections, and TRAP and hematoxylin-and-eosin (H&E) staining analyses were performed, as previously mentioned [2]. TRAP-positive cells were counted using Image J (Media Cybernetics, Inc. Rockville, MD, USA) [33].

To identify proinflammatory cytokine expressions, IHC analysis was performed with primary antibodies against IL-1β (AF-401-NA, R&D Systems, Minneapolis, MN, USA), IL-6 (ab6672, Abcam, UK), and TNF-a (sc-133192, Santa Cruz, CA, USA). For the secondary antibody reactions, VECTASTAIN^®^ Universal Quick Kit (PK-7800, Vector Laboratories, Burlingame, CA, USA) for IL-1β and Dako EnVision+ System Kit (K4009, Dako, Denmark) for IL-6 and TNF-a were used. The DAB Peroxidase Substrate Kit was used to visualize the cytokine expressions in the tissues [2]. The quantitative comparisons were conducted by calculating the percentage of positive staining on the same-sized area in the furcation of the maxillary second molar using Image J Fiji (Version 1.2).

### 4.6. Primary Cell Isolation and Cultures

BMSCs and BMMs were isolated from C57BL/6 mice (5–8 weeks old) femurs and tibias, as previously described (Zang et al., 2020; Song et al., 2022), and maintained in GM. In case of BMM cultures, the cells were cultured in the GM including 100 ng/mL of mouse macrophage colony-stimulating factor (M-CSF, BioLegend, San Diego, CA, USA) for 3 days, and then treated with M-CSF (30 ng/mL) and RANKL (100 ng/mL, Peprotech, Inc., Seoul, Republic of Korea) to induce osteoclast differentiation. For inflammatory cytokine expression, LPS (100 ng/mL, *Escherichia coli O111:B4*, Sigma Aldrich, St. Louis, MO, USA) was treated for 15 min or 6 h. Various concentrations of PFD were added, as shown in figures.

For osteoblast differentiation, BMSCs were incubated in the OM, including L-ascorbic acid (A.A., 50 µg/mL, Sigma Aldrich, St. Louis, MO, USA), β-glycerophosphate (β-GP, 5 mM, ChemCruzTM biochemiacls, Dallas, TX, USA), and bone morphogenetic protein 2 (BMP2, 100 ng/mL, Cowellmedi Co., Ltd., Seoul, Republic of Korea). All cell culture media were changed every other day.

### 4.7. TRAP staining, Filamentous (F)-Actin Ring Formation, and Resorption Pits Assays

For in vitro TRAP staining, BMMs were cultured with PFD (0, 100, 200, 400, 800 µM) in the presence of M-CSF (30 ng/mL, BioLegend, San Diego, CA, USA) and RANKL (100 ng/mL, Peprotech, Inc., London, UK) for 3–4 days. Thereafter, cells were fixed in 10% formalin for 15 min; then, they were infiltrated in TRAP staining solution (Cosmo Bio Co., LTD. Tokyo, Japan) at 37 °C for 30 min. The TRAP+ cells (≥3 nuclei) with distinct stains were identified as osteoclasts, and the number of osteoclasts was acquired under microscopy (DMIL LED/DFC450C, Leica, Germany).

For F-actin ring staining, the cells were fixed as aforementioned. After being permeated with 0.1% Triton X-100, the cells were stained with Actin-Stain™ Fluorescent Phalloidins (Red, Thermo Fisher, Pleasanton, CA, USA) for filamentous actin ring belt and 4′, 6-diamidino-2-phenylindole (DAPI, blue, Cell Signaling Technology, Danvers, MA, USA) for nucleus. Fluorescence images were obtained via a microscope (DMIL LED/DFC450C, Leica, Germany).

Resorption pit assay was performed in BMM cultures. Cells were seeded on bone slices in the presence of M-CSF (30 ng/mL) and RANKL (50 ng/mL), with or without PFD for 12 days. After reacting with 5% sodium hypochlorite (Sigma Aldrich, St. Louis, MO, USA) for 3–5 min, the bone slices were stained with toluidine blue (1 mg/mL, St. Louis, MO, USA). Images were obtained under microscopy (DMIL LED/DFC450C, Leica, Germany). The quantitative analysis was performed using Image J.

### 4.8. Immunofluorescence Experiment

BMMs were fixed in 10% formalin solution for 15 min before being blocked in PBS with 5% bovine serum albumin (BSA, St. Louis, MO, USA), reacted with 3% Triton X-100 for 60 min, and then incubated with antibody for p65 (1:200, #6956, Cell Signaling Technology, Danvers, MA, USA) overnight at 4 °C. Thereafter, cells were incubated with corresponding second antibody (Alexa Fluor 488, 1:1000, Invitrogen, Carlsbad, CA, USA) and counterstained with DAPI for nuclei in 5 min. The images were photographed using microscopy (Lionheart FX Automated Microscope, Agilent, CA, USA).

### 4.9. Immunoassays

The proteins were isolated from cultured cells using a cell lysis buffer (Cell Signaling Technology, Danvers, MA, USA). The nuclear and cytoplasmic proteins were isolated from cultured cells using the NE-PER Nuclear and Cytoplasmic Ex-traction Reagents Kit (Thermo Fisher, Pleasanton, CA, USA), according to the manufacturer’s instructions. After the samples were centrifuged for 15 min at 13,200 rpm in 4 °C, the liquid supernatant was quantified using a protein assay reagent (Bio-Rad Laboratories, Hercules, CA, USA). Then, the protein reagents were loaded in 10–14% SDS-PAGE and transferred to 0.2 µm of PVDF membranes (Amersham^TM^, Little Chalfont, UK). After shaking for 35 min in 5% skim milk in TBS containing 0.1% Tween 20 (TBST), the membranes were reacted slowly in a primary antibody solution (5% skim milk in TBST containing different primary antibodies) overnight at 4 °C and then incubated with the corresponding secondary antibodies (1:10,000) for 1 h at room temperature. After being washed with TBST thrice, the membranes were detected using a western chemiluminescent HPR substrate (Millipore, Burlington, MA, USA) and the Azure Western blot imaging system (Azure 600, Azure Biosystems, Dublin, CA, USA). The primary antibodies are as follows: NFATc1 (sc-7294), CtsK (sc-48353), Lamin B (sc-374015), IκBa (sc-371), and β-Actin (sc-47778) were provided by Santa Cruz. IL-1β (12507S), IKKβ (2678S), phospho-IKKa/β (2078), p65 (4764), phospho-p65 (3033S), phospho-IκBa (2859S), c-Fos (2250), p38 (9212S), phospho-p38 (9211S), ERK (9102S), phospho-ERK (9101S), JNK (9258S), phospho-JNK (9251S), and MEK2 (9125S) were purchased from Cell Signaling Technology. IL-6 (ab6672) was purchased from Abcam. All concentrations of the primary antibodies were based on instruction manuals of antibody manufacturing companies.

### 4.10. RT-PCR Analysis

The total RNA was extracted from samples using a TRIzol reagent (Invitrogen, Waltham, MA, USA). The RNA concentrations were assessed via the µDropTM Duo Plate (Thermo Fisher, Pleasanton, CA, USA). Afterwards, reverse transcription was performed using dNTP (0.5 mM), random primers (0.2 µg), RNAsin (40 U), and MMLV (200 U) (Promega, Madison, WI, USA) according to the manufacturer’s instructions.

The Power SYBR green PCR master mix (Life Technologies LTD, Woolston Warrington, UK) was used for real-time PCR performed using StepOnePlus™ Real-Time PCR System (Applied Biosystems, Foster City, CA, USA). RT-PCR was performed at 95 °C (10 min) in the hold stage, followed by 40 cycles at 95 °C (30 s), 55 °C (30 s), and 72 °C (30 s) in the PCR stage and at 95 °C (15 s), 60 °C (60 s), and 95 °C (15 s) in the melt-curve stage. The 18S RNA gene was defined as the endogenous gene for normalization. The comparative Ct method was used to quantitatively analyze relative target gene expressions in StepOne Software v2.1 (Applied Biosystems, Waltham, MA, USA).

The sequences of primers used for real-time PCR were as follows: for *18s*, 5′-GGC CGT TCT TAG TTG GTG GA-3′ and 5′-CCC GGA CAT CTA AGG GCA TC-3′; for *Il-1β*, 5′-GCA CTA CAG GCT CCG AGA TGA AC-3′ and 5′-TTG TCG TTG CTT GGT TCT CCT TGT-3′; for *Il-6*, 5′-CTT CCA TCC AGT TGC CTT CT-3′ and 5′-AAT TAA GCC TCC GAC TTG TG-3′; for *Tnf-a*, 5′-CGT CGT AGC AAA CCA CCA AG-3′ and 5′-GAG ATA GCA AAT CGG CTG ACG-3′; for *c-Fos*, 5′-GGG AAT GGT GAA GAC CGT GT-3′ and 5′-GCA ATC TCA GTC TGC AAC GC-3′; for *Nfatc1*, 5′-CTC GAA AGA CAG CAC TGG AGC AT-3′ and 5′-CGG CTG CCT TCC GTC TCA TAG-3′; for *Ctsk*, 5′-TAC CCA TAT GTG GGC CAG GA-3′ and 5′-ATA GCC CAC CAC CAA CAC TG-3′; for *Trap*, 5′-TCC GTG CTC GGC GAT GGA CCA GA-3′ and 5′-CTG GAG TGC ACG ATG CCA GCG ACA-3′; for *Mmp9*, 5′-TGG TCT TCC CCA AAG ACC TG-3′ and 5′-AGG TTT GGA ATC GAC CCA CG-3′; for *Alp*, 5′-TAC ATT CCC CAT GTG ATG GC-3′ and 5′-ACC TCT CCC TTG AGT GTG GG-3′; for *Osx*, 5′-AGC GAC CAC TTG AGC AAA-3′ and 5′-GCG GCT GAT TGG CTT CTT CT-3′; for *Runx2*, 5′-TCT CCA ACC CAC GAA TGC ACT A-3′, and 5′-ATA GCG TGC TGC CAT TCG AGG T-3′.

### 4.11. ARS

BMSCs were cultured with GM or OM with PFD (0, 100, 200, 400, 800 µM) for 14 days, fixed using 10% formalin for 15 min, and treated with ARS solution (40 mM, pH 4.2, Sigma Aldrich, St. Louis, MO, USA) for 15 min. For quantification, the ARS-stained calcium deposits were extracted for 20 min using sodium phosphate solution (10 mM, pH 7.0), including 10% cetylpyridinium chloride (Sigma Aldrich, St. Louis, MO, USA). Once comprehensive dissolution was attained, the absorbance was measured at a wavelength of 562 nm on a microplate reader (Multiskan SkyHigh Micro-plate Reader, Thermo Fisher, Pleasanton, CA, USA).

### 4.12. Statistical Analysis

Western blot analysis was repeated three times using individually cultured cells. Other in vitro experiments were performed once in triplicate. For in vivo experiments, the total number of mice used is indicated in the figure legends. Data are expressed as means ± standard deviations (SDs), and statistical analysis was performed using one-way ANOVA and Tukey’s honestly using GraphPad Prism 9 (GraphPad Software Inc, San Diego, CA, USA). *p*-values were expressed as follows: **p* < 0.05, ** *p* < 0.01, and *** *p* < 0.001.

## 5. Conclusions

PFD inhibits alveolar bone loss and the expression of IL-1β, IL-6, and TNF-a in ligature-induced periodontitis in mice. Its protective effects are due to the inhibition of osteoclast differentiation and inflammation-related cytokine expression by suppressing the activation of the NF-κB pathway. Our results indicate that PFD can be used as an adjunctive therapy for periodontal disease in combination with antibiotics.

## Figures and Tables

**Figure 1 ijms-24-08682-f001:**
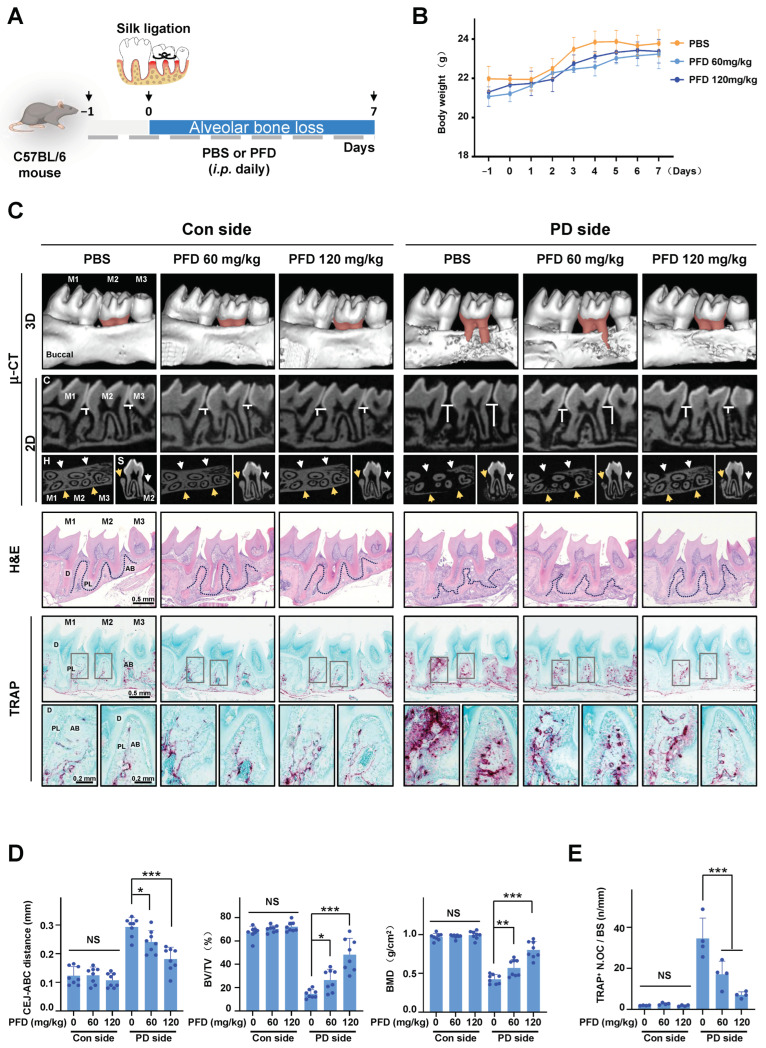
PFD inhibits alveolar bone loss in ligature-induced periodontitis in mice. (**A**) Establishment of mice periodontitis model and schedule of PFD treatment in schematic illustration. (**B**) Changes in body weight. *n* = 8. (**C**) Representative μ-CT images. The areas of CEJ-ABC in the second molars are presented in red color. The distances of CEJ-ABC around the second molars are shown in 2D coronal (**C**) images. The yellow (buccal) and white (lingual) arrows indicate the typical part of alveolar bone around the second molar in the 2D horizontal (H) and sagittal (S) images. H&E staining and TRAP staining in the longitudinal sections of tooth and maxillary tissues are different among groups after PFD intervention. Scale bar, 0.5 mm. The dotted lines in the H&E staining images mark the margin of alveolar bone around the second molar. The last line images of (**C**) are the typical local amplification of TRAP staining; the corresponding amplifying parts are marked with gray squares in TRAP staining images. Scale bar, 0.2 mm. The TRAP + cells are presented in purple. (**D**) Quantitative analysis of CEJ-ABC distance. BV/TV and BMD are shown at the bone surrounding the second molar. *n* = 8. (**E**) The number of TRAP + cells in tissues was counted. *n* = 4. Pirfenidone, PFD; i.p. intraperitoneal injection; M1, first molar; M2, second molar; M3, third molar; Con, control; PD, periodontitis; μ-CT, micro-computed tomography; H&E, hematoxylin and eosin staining; TRAP, TRAP staining; C, coronal; H, horizontal; S sagittal; CEJ-ABC, cementoenamel junction to alveolar bone crest; BMD, bone mineral density; BV/TV, bone volume per tissue volume; D, dentin; PL, periodontal ligament; AB, alveolar bone; TRAP+ N.OC/BS, TRAP-positive number of osteoclast per bone surface. All bar graphs are showed as the mean ± SD. * *p* < 0.05, ** *p* < 0.01, and *** *p* < 0.001 compared with the PBS group in periodontitis side; NS, non-significant compared with the PBS group in control side.

**Figure 2 ijms-24-08682-f002:**
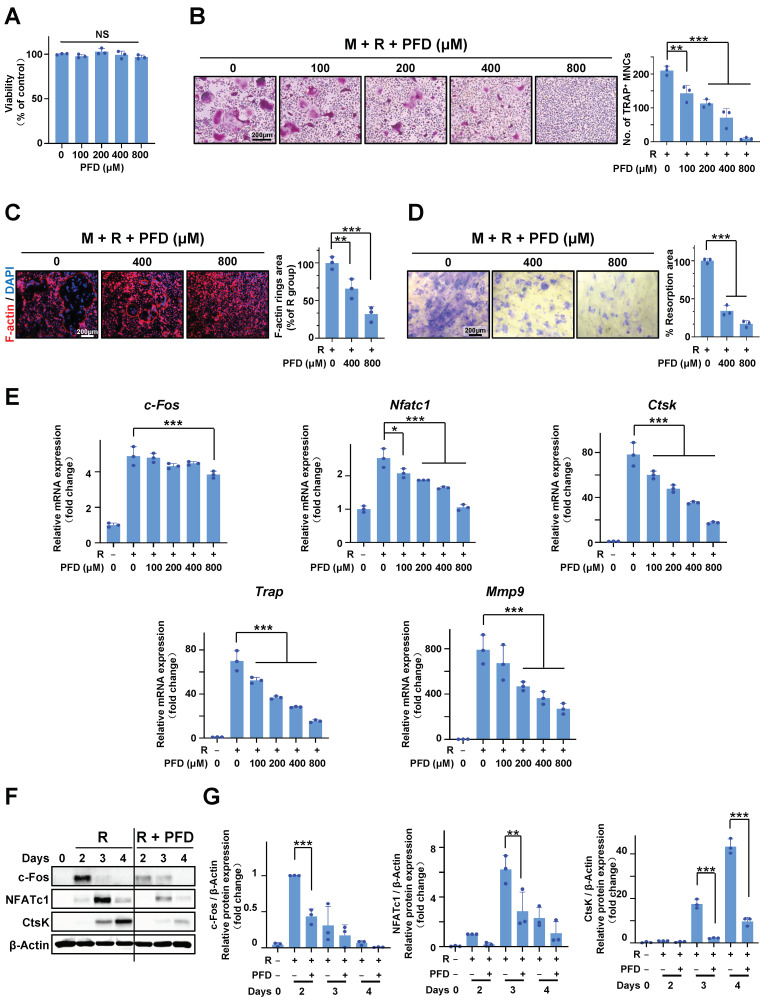
PFD suppresses RANKL-induced osteoclast differentiation in BMMs. (**A**) The MTT assay of BMMs with PFD (0, 100, 200, 400, 800 μM) treatment for 1 day. (**B**–**G**) BMMs were cultured with 30 ng/mL M-CSF (M) and 100 ng/mL RANKL (R) containing different concentrations of PFD for 3–5 days except for special notes. (**B**) TRAP staining images (left panel) and quantification of the TRAP+ cells (nuclei > 3, right graph). (**C**) F-actin belt (red) and nuclei (DAPI, blue) staining images of osteoclasts (left panel) and quantification of F-actin belt area (right graph). (**D**) Pit formation assay. BMMs were seeded on bone slices and cultured with 30 ng/mL MCSF and 50 ng/mL RANKL together with PFD (0, 400, 800 μM) for 12 days. A representative image (left panel) and quantification of the pit formation area (right graph). (**E**) The gene expression of *c-Fos*, *Nfatc1*, *Ctsk*, *Trap*, and *Mmp9* was quantified via RT-PCR. (**F**) Representative Western blot images of c-Fos, NFATc1, and CtsK expression. (**G**) A relative gray densitometric analysis of (**F**). The fold value was based on RANKL group cultured for two days. All quantifications of images were measured in Image J. Scale bar, 200 µm. Pirfenidone, PFD; M, MCSF; R, RANKL; No., number; TRAP + MNCs, TRAP-positive multi-nuclei cells. All bar graphs are showed as the mean ± SD. * *p* < 0.05, ** *p* < 0.01, and *** *p* < 0.001 compared with the RANKL group; *n* = 3. NS, non-significant compared with the untreated-PFD group.

**Figure 4 ijms-24-08682-f004:**
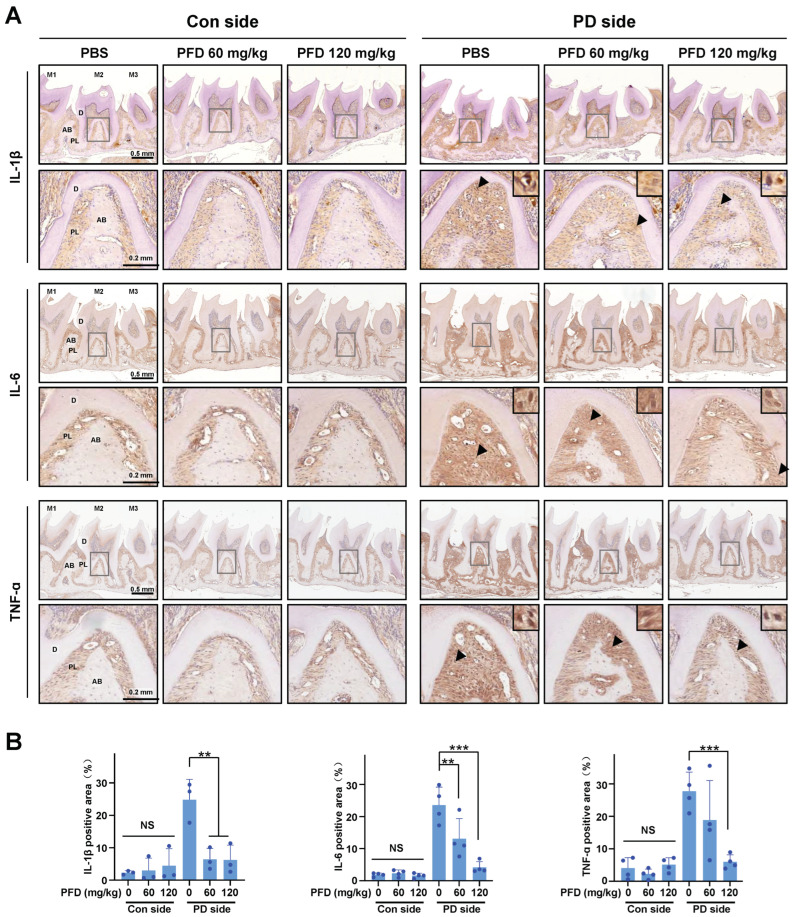
PFD decreases proinflammatory cytokine expression in ligature-induced periodontitis. (**A**) IHC staining images of IL-1β, IL-6, and TNF-a in the longitudinal section of tooth and maxillary tissues in different groups after PFD intervention. Scale bar, 0.5 mm. The second-line images of each IHC staining are typical local magnification, and the corresponding magnifying parts are marked with gray squares in IHC staining images. Black arrowhead: IL-1β, IL-6, and TNF-a positive cells were enlarged and shown at top right corner. Scale bar, 0.2 mm. (**B**) Comparisons of the IL-1β, IL-6, and TNF-a-positive area in the root furcation area of the second molar. Pirfenidone, PFD; M1, first molar; M2, second molar; M3, third molar; Con, control; PD, periodontitis; D, dentin; PL, periodontal ligament; AB, alveolar bone. All bar graphs are showed as the mean ± SD. ** *p* < 0.01, and *** *p* < 0.001 compared with the PBS group in periodontitis side; NS, non-significant compared with the PBS group in control side. *n* = 3.

**Figure 5 ijms-24-08682-f005:**
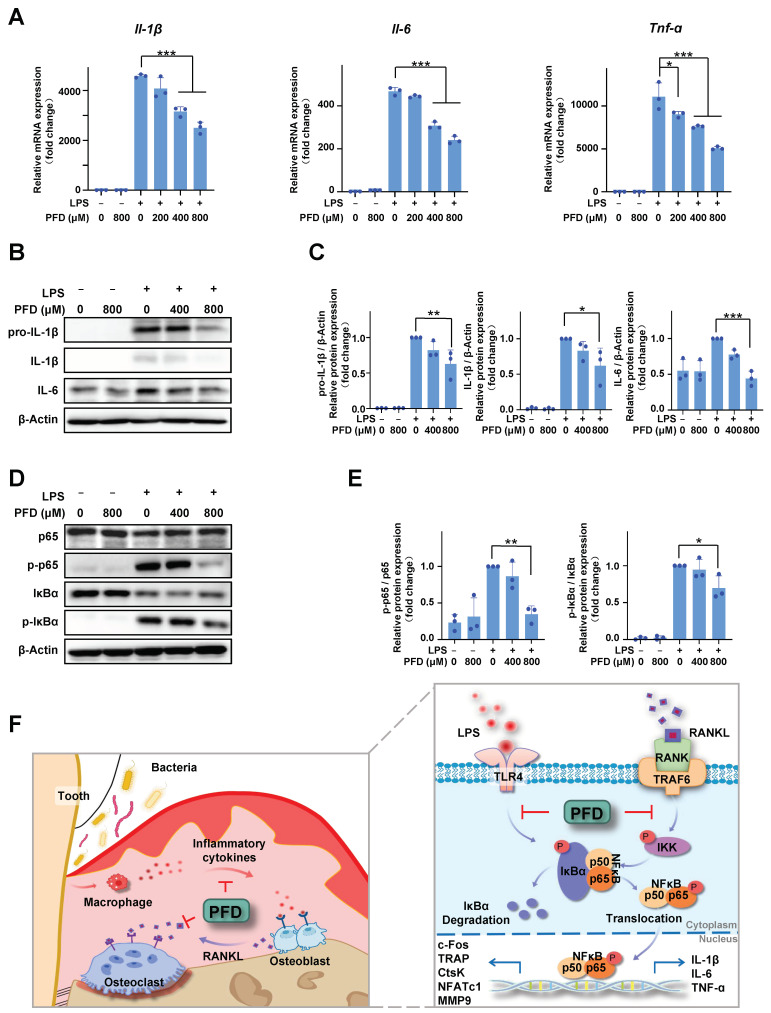
PFD inhibits LPS-induced proinflammatory cytokine expression and activation of the NF-κB pathway in BMMs. (**A**–**E**) BMMs were treated with LPS (100 ng/mL) after pretreatment with PFD for 30 min. (**A**) BMMs were incubated in various concentrations of PFD with or without 100 ng/mL LPS for 6 h, and RT-PCR was performed. (**B**,**C**) Western blot analysis for pro-IL-1β, IL-1β, and IL-6 expression (**D**,**E**) Western blot analysis for p65, p-p65, IκBa, and p-IκBa expression. (**C**) and (**E**) are the relative gray densitometric analysis of (**B**) and (**D**), respectively. The fold value was based on RANKL group. All quantifications of images were measured in Image J from three independent experiments, respectively. (**F**) Schematic representation of PFD mechanism model in periodontitis. PFD inhibits ligature-induced inflammatory cytokine expression and osteoclastogenesis via suppressing NF-κB activation. Pirfenidone, PFD; All bar graphs are showed as the mean ± SD. * *p* < 0.05, ** *p* < 0.01, and *** *p* < 0.001 compared with the LPS group. *n* = 3.

## Data Availability

The raw data supporting the findings of this study will be made available by the authors, without undue reservation, once the article is accepted.

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
