# Peer review of "Pirfenidone Inhibits Alveolar Bone Loss in Ligature-Induced Periodontitis by Suppressing the NF-κB Signaling Pathway in Mice"

_ijms, 2023, doi:10.3390/ijms24108682_

Round 1
Reviewer 1 Report
The manuscript by Zhang et al. explores the therapeutic potential of the FDA-approved drug Pirfenidone (PFD) on experimental periodontitis in mice. The study demonstrated that PFD reduces the bone destruction and inflammatory response associated with the ligature-induced PD model. Also, using mice BMM, the authors showed the PFD-dose-dependent suppression of osteoclasts differentiation, inhibition of the NFkb pathway, p65 nuclear translocation, and lastly, LPS-induced expression of inflammatory cytokines. Overall the manuscript is well structured, and the data support the conclusions. I have minor comments on the study above as follows:
-All experiments were performed once in triplicate; how many technical and biological replicas were used? In all figures, the n=3 (a minimal sample size), but in Figure 4B, four individual samples are in the graph.
Why did the authors use E coli LPS to treat the BMM when the ligature-associated bacteria were available to have a bacteria-specific response? Or, even more closely related to humans, they could have used Pg LPS.
Line 36, please refer to the current classification scheme for periodontal and peri-implant diseases and conditions. PD is not defined as a bacterial infection of the oral cavity but a dysbiotic microbiome-driven chronic inflammatory disease.
Line 38, periodontal treatment has surgical and non-surgical approaches based on eliminating the pathogenic biofilm rather than tissue.
Line 39, what do the authors mean by "precisely control the periodontitis"? Is it antigen-specific?
Minor grammar mistakes, i.e., line 17, "controlling the periodontitis," line 21, "the micro-computed tomography and histology analyses were performed," Line 161 should be the nuclear translocation of p65. Line 65, with high stability?
Reviewer 2 Report
I was pleased to review the manuscript “ijms-2337813” entitled “Pirfenidone inhibits alveolar bone loss in ligature-induced periodontitis by suppressing the NF-κB signaling pathway in mice” for International Journal of Molecular Sciences. The study aimed to investigate the effects of PFD on alveolar bone loss in ligature-induced periodontitis in mice and to identify the mechanisms involved. The experimental periodontitis was established in mice by ligating the unilateral maxillary second molar for seven days, and PFD was administered daily by intraperitoneal injection. Micro-computed tomography and histology analyses were performed to evaluate the effects of PFD treatment. In vitro analysis using bone marrow macrophages (BMMs) showed that PFD inhibited osteoclast differentiation and proinflammatory cytokine expression by suppressing the NF-κB signal pathway. The manuscript shows an adequate design with robust methodologies. However, there is space to be improved, especially in the methodology section.
The whole manuscript should be carefully read to correct several typos.
The figure panel 4 should be amended to include images where the readers can see the immunolabeling. As it is, the cells pointed by the black arrowheads are indistinguishable.
The discussion MUST be improved.
What are the side effects of PFD?
The second paragraph should be rewritten to better discuss the findings instead of citing the results again.
How LPS activates RANKL/RANK/TRAF6?
Some parts need to be clarified where the authors use too many IFD concepts to discuss their results.
METHODS
In vivo and In vitro should be italicized;
Correct the Sigma Aldrich typo;
Correct seeding density formatting;
What growth medium was used?
What MTT reagent?
How were 8 animals divided into 3 groups with n=8?
Improve animal experiment surgery description.
Why the author chose 60 mg/kg or 120 mg/kg of PFD?
What are the secondary antibodies for IHC? How was the IHC revealed? What parameters were used to quantify the IHC labeling?
Round 2
Reviewer 2 Report
The authors substantially improved the manuscript, which now looks suitable to be published.